

# Tobacco rattle virus-induced *PHYTOENE DESATURASE* (*PDS*) and *Mg-chelatase H subunit* (*ChlH*) gene silencing in *Solanum pseudocapsicum* L.

Hua Xu[1,2], Leifeng Xu[1], Panpan Yang[1,3], Yuwei Cao[1], Yuchao Tang[1], Guoren He[1], Suxia Yuan[1] and Jun Ming[1]

[1] Institute of Vegetables and Flowers, Chinese Academy of Agricultural Sciences, Beijing, China
[2] College of Life and Environmental Science, GanNan Normal University, Ganzhou, China
[3] College of Landscape Architecture, Nanjing Forestry University, Nanjing, China

## ABSTRACT

Virus-induced gene silencing (VIGS) is an attractive tool for determining gene function in plants. The present study constitutes the first application of VIGS in *S. pseudocapsicum*, which has great ornamental and pharmaceutical value, using *tobacco rattle virus* (TRV) vectors. Two marker genes, *PHYTOENE DESATURASE* (*PDS*) and *Mg-chelatase H subunit* (*ChlH*), were used to test the VIGS system in *S. pseudocapsicum*. The photobleaching and yellow-leaf phenotypes of the silenced plants were shown to significantly correlate with the down-regulation of endogenous *SpPDS* and *SpChlH*, respectively ($P \leq 0.05$). Moreover, the parameters potentially affecting the efficiency of VIGS in *S. pseudocapsicum*, including the Agrobacterium strain and the inoculation method (leaf syringe-infiltration, sprout vacuum-infiltration and seed vacuum-infiltration), were compared. The optimized VIGS parameters were the leaf syringe-infiltration method, the Agrobacterium strain GV3101 and the growth of agro-inoculated plants at 25°. With these parameters, the silencing efficiency of *SpPDS* and *SpChlH* could reach approximately 50% in *S. pseudocapsicum*. Additionally, the suitability of various reference genes was screened by RT-qPCR using three candidate genes, and the results demonstrated that glyceraldehyde 3-phosphate dehydrogenase (*GAPDH*) can serve as a suitable reference for assessing the gene expression levels of VIGS systems in *S. pseudocapsicum*. The proven application of VIGS in *S. pseudocapsicum* and the characterization of a suitable reference gene in the present work will expedite the functional characterization of novel genes in *S. pseudocapsicum*.

## INTRODUCTION

*Solanum pseudocapsicum* L., which belongs to the genus *Solanum* and the family Solanaceae, possesses high ornamental value and is widely cultivated as an indoor ornament due to its bright red berries at maturity (*Aliero, Grierson & Afolayan, 2006b*). This plant also has important pharmaceutical value, as reflected by its use for the treatment of boils,

Corresponding author
Jun Ming, mingjun@caas.cn

gonorrhoea and acute abdominal pain (*Aliero, Grierson & Afolayan, 2006a*). The total alkaloid fraction from *S. pseudocapsicum* has hepatoprotective properties and antitumour, antifungal, and antimicrobial activities (*Mitscher, Juvarkar & Beal, 1976*; *Badami et al., 2003*; *Vijayan et al., 2003*; *Aliero, Grierson & Afolayan, 2006b*). An ethyl acetate extract from *S. pseudocapsicum* showed promising antifeedant and insecticidal activities (*Jeyasankar, Premalatha & Elumalai, 2012*). Despite the great ornamental and pharmaceutical value of *S. pseudocapsicum*, scarce molecular-level information is currently available due to the lack of gene function identification techniques and the unavailability of related genome sequences. Gene function studies would yield a better understanding of the molecular mechanisms that regulate trait development in *S. pseudocapsicum* and could provide a gene source for molecular breeding and drug development. Consequently, the development of techniques for identifying the functions of genes in *S. pseudocapsicum* is vital.

Virus-induced gene silencing (VIGS) is a recently developed gene transcript suppression technique for characterizing gene function in plants (*Burch-Smith et al., 2004*). This technique has many advantages for unravelling gene function compared with the loss-of-function approaches that are currently available for plants, and these include easy and rapid gene silencing, its transformation-independent nature, and the requirement of only some sequence information (*Burch-Smith et al., 2004*). Increasing numbers of viruses have recently been used to derive VIGS vectors for unravelling gene functions, such as *tobacco rattle virus* (TRV) (*Ratcliff, Martin-Hernandez & Baulcombe, 2001*), *barley stripe mosaic virus* (BSMV) (*Holzberg et al., 2002*), *cabbage leaf curl virus* (CaLCuV) (*Muangsan et al., 2004*), *potato virus X* (PVX) (*Faivre-Rampant et al., 2004*) and *cucumber mosaic virus* (CMV) (*Tasaki et al., 2016*). Among these viral vectors, TRV has been widely used to construct VIGS vectors for silencing target genes in various plant species, such as *Arabidopsis thaliana* (*Burch-Smith et al., 2006*), *Nicotiana benthamiana*, *Solanum lycopersicum* (*Liu, Schiff & Dinesh-Kumar, 2002*), and *Capsicum annuum* (*Chung et al., 2004*), because it has a wide host range and induces mild infection symptoms. However, it is unknown whether TRV-based VIGS can be applied to unravel gene functions in *S. pseudocapsicum*. Although the sequence of the *S. pseudocapsicum* genome is not currently available, the genome sequences of other species belonging to the Solanum genus (Solanaceae), such as *S. lycopersicum* and *C. annuum* (*Lin et al., 2014*; *Qin et al., 2014*), have been published. Thus, the target gene sequence of *S. pseudocapsicum* can be obtained through homology-based cloning, making the identification of gene functions in *S. pseudocapsicum* possible. *PDS* encodes an important enzyme in carotenoid biosynthesis, and the *ChlH* gene encodes the H subunit of magnesium-protoporphyrin chelatase, which is involved in chlorophyll biosynthesis. Both the *PDS* and *ChlH* genes are commonly used as markers of VIGS because the silencing of one of these genes yields a photobleached and a yellow-leaf phenotype, respectively (*Cunningham & Gantt, 1998*; *Hiriart et al., 2002*; *Liu et al., 2012*). To assess whether the TRV-based VIGS system could be used to identify gene functions in *S. pseudocapsicum*, we investigated a TRV-induced *PDS* and *ChlH* silencing protocol involving agro-infiltration in *S. pseudocapsicum*. Moreover, the effects of the Agrobacterium strain, the inoculation methods and the growth temperature after Agrobacterium infiltration on *S. pseudocapsicum* were compared.

Reverse transcription quantitative real-time PCR (RT-qPCR) is a standard tool for the quantification of gene expression levels. In the present study, RT-qPCR was used to detect the *PDS* and *ChlH* gene expression levels. It is well known that the accuracy of RT-qPCR techniques is challenged by many sources of variation, including template quality, sampling errors, and amplification efficiency (*Bustin, 2002*). The selection of a suitable reference gene is vital in RT-qPCR analysis because it ensures the reliability of the RT-qPCR results (*Vandesompele et al., 2002*). Housekeeping genes are commonly used as reference genes for normalizing RT-qPCR analyses in molecular biology (*Hong et al., 2008*). For example, genes encoding actin (*ACTIN*), polyubiquitin (*UBQ*), and glyceraldehyde-3-phosphate dehydrogenase (*GAPDH*) are widely used as reference genes in gene expression studies (*Czechowski et al., 2005*; *Jain et al., 2006*). To date, a reference gene in *S. pseudocapsicum* has not been identified. Statistical algorithms such as geNorm, BestKeeper and NormFinder have been developed to evaluate the best-suited reference genes for normalizing real-time PCR data from a given set of biological samples (*Vandesompele et al., 2002*; *Andersen, Jensen & Orntoft, 2004*; *Pfaffl et al., 2004*). Thus, three reference genes (*ACTIN*, *GAPDH* and *UBQ*) identified in previous studies were selected as candidates in the present study for stability screening using the geNorm, BestKeeper and NormFinder algorithms prior to VIGS.

This study established a TRV-based VIGS protocol and provided the first screening of appropriate reference genes for *S. pseudocapsicum*. The results will expedite the characterization of gene expression levels and the analysis of gene functions in *S. pseudocapsicum*, which would allow the exploitation of desirable genes in *S. pseudocapsicum*.

## MATERIALS AND METHODS

### Plant materials

Seeds from *S. pseudocapsicum* were soaked in water at 25 °C for five days and then sown in pots containing an autoclaved medium consisting of peat and vermiculite (at a ratio of 3:1). The seedlings were grown in a growth chamber (25 °C, 60–70% relative humidity, 16-h light/8-h dark cycle).

For the VIGS experiments, leaves exhibiting the silencing phenotype (only the phenotypic areas) were collected from five of the silenced seedlings, and leaves were also harvested from five mock-treated and untreated seedlings at the second-true-leaf stage. The collected leaves were used for analysis of the expression of *PDS* and *ChlH* genes.

For the screening of reference genes, leaves at different development stages (cotyledons stage, first-true-leaf stage and second-true-leaf stage), stems and roots were sampled to analyse the expression of the candidate reference genes. Samples were obtained from five different plants, three independent biological replicates were included for each sample, and each experiment was repeated three times. All the samples were immediately frozen in liquid nitrogen and maintained at −80 °C until RNA extraction.

## RNA extraction and RT-PCR

Total RNA was extracted using an RNAprep Pure Plant Kit (Polysaccharide-& Polyphenolic-rich) (TIANGEN, Beijing, China) according to the manufacturer's instructions, and any DNA contamination was removed with RNase-free DNase I (TIANGEN, Beijing, China). The RNA integrity was confirmed by denaturing 1.2% agarose gel electrophoresis, and the quantity and purity were determined with a Q3000 UV spectrophotometer (Quawell Technology Inc., San Jose, CA, USA). Only high-quality samples with an A260/A230 > 2.0 and 1.8 < A260/A280 < 2.1 were used for cDNA synthesis. First-strand cDNA was synthesized using 1 μg of total RNA using the TransScript® One-Step gDNA Removal and cDNA Synthesis SuperMix Kit (TRAN, Beijing, China).

## Cloning of the candidate reference genes and the *PDS* and *ChlH* genes

To acquire the *S. pseudocapsicum* orthologues of the reference genes as well as *PDS* and *ChlH* genes that were previously reported in other plant species, publicly available gene fragments, including *ACTIN, GAPDH, UBQ, PDS* and *ChlH*, from species in Solanaceae (tomato, pepper, tobacco and potato) were downloaded. The *ACTIN, GAPDH, UBQ, PDS* and *ChlH* gene fragments in *S. pseudocapsicum* were amplified by PCR using homology-based cloning as previously reported (*Zhong et al., 2014*). Primers for the cloning of *SpACTIN* were designed according to the conserved regions among *S. lycopersicum* (*SlACTIN*, NM_001330119.1), *C. annuum* (*CaACTIN*, XM_016683691.1), *Solanum tuberosum* (*StACTIN*, XM_015308091.1) and *N. tabacum* (*NtACTIN*, EU938079.1); primers for the cloning of *SpGAPDH* were designed according to the conserved regions among *S. tuberosum* (*StGAPDH*, XM_006352526.2), *C. annuum* (*CaGAPDH*, NM_001324619.1), *N. tabacum* (*NtGAPDH*, XM_016612593.1) and *S. lycopersicum* (*SlGAPDH*, XM_004248266.3); primers for the cloning of *SpUBQ* were designed according to the conserved regions among *C. annuum* (*CaUBQ*, AY486137.1), *S. tuberosum* (*StUBQ*, XM_015307154.1), *S. lycopersicum* (*SlUBQ*, XM_019214130.1) and *N. tabacum* (*NtUBQ*, XM_016589747.1); primers for the cloning of *SpPDS* were designed according to the conserved regions among *S. lycopersicum* (*SlPDS*, EF650011.1), *C. annuum* (*CaPDS*, NM_001324813.1) and *N. tabacum* (*NtPDS*, DQ469932.1); and primers for the cloning of *SpChlH* were designed according to the conserved regions among *N. tabacum* (*NtChlH*, NM_001325713.1), *S. lycopersicum* (*SlChlH*, XM_015217369.1) and *C. annuum* (*CaChlH*, XM_016716472.1). RT-PCR was performed using the cDNA of leaves of *S. pseudocapsicum* as the template. The RT-PCR temperature protocol was as follows: (1) 95 °C for 5 min, (2) 35 cycles of 95 °C for 30 s, 55 °C for 30 s and 72 °C for 1 min, and (3) maintenance at 72 °C for 2 min. To obtain the longer *PDS* gene fragment, the *PDS* gene sequence was further cloned by 3′-rapid amplification of cDNA ends (3′-RACE) using the GeneRacer™ kit (Invitrogen, Carlsbad, CA, USA). The gene-specific primers (GSPs) for 3′-RACE were designed based on the sequence of the *PDS* fragment amplified using homology-based cloning according to the manufacturer's instructions provided with the GeneRacerTM kit. The reaction programmes were as followers: (1) 95 °C for 3 min, (2) 35 cycles of 98 °C

**Table 1  RT-qPCR primers used in the analyses.**

| Gene symbol | Primer name | Primer sequence (5′–3′) | Product length (bp) | $R^2$ | E% |
|---|---|---|---|---|---|
| ACTIN | Sp-YG-ACT-F<br>Sp-YG-ACT-R | ATTGAGCATGGCATTGTGAGC<br>GCGATTAGCCTTTGGATTGAGA | 137 | 0.982 | 97.7 |
| GAPDH | Sp-YG-GAPDH-F<br>Sp-YG-GAPDH-R | CCAACCCTTGTCTTCCCACC<br>CTCAAACCTACCGCCTCCCT | 242 | 0.990 | 103.8 |
| UBQ | Sp-YG-UBQ-F<br>Sp-YG-UBQ-R | TTGGCAAGCAACAATCAT<br>GCAGATGGACAGCAGGAC | 225 | 0.986 | 105.7 |
| PDS | Sp-YG-PDS-F<br>Sp-YG-PDS-R | TCATGTTGTCAAAACTCCAAGG<br>TGTCAACTTCTTCTCGCTCC | 223 | 0.995 | 96.7 |
| ChlH | Sp-YG-ChlH-F<br>Sp-YG-ChlH-R | AAGCACCTGGTAATCTGAACTCTG<br>CATCGGGTCACCTTCGTATC | 112 | 0.991 | 97.1 |

for 20 s, 58 °C for 15 s and 72 °C for 2 min, and (3) maintenance at 72 °C for 2 min. The above-mentioned primer sequences are shown in Table S1.

## Analysis of gene expression by RT-qPCR and detection of TRV by RT-PCR

Real-time PCR was performed using SYBR Premix Ex Taq (Takara, Japan). The primers used to amplify the *ACTIN, GAPDH, UBQ, PDS* and *ChlH* segments are shown in Table 1. The reaction programmes were as followed: (1) 95 °C for 1 min, (2) 40 cycles of 95 °C for 20 s, 60 °C for 10 s, and 72 °C for 25 s, and (3) a melt curve programme (65 °C to 95 °C with an increment in temperature of 0.5 °C every 0.05 s). The signals were monitored using a CFX96 Real-Time System (Bio-Rad, Hercules, CA, USA). For each primer pair, the amplification efficiency was derived from a standard curve generated from five-fold serial dilution points of a mixture of the cDNA from all the samples. The expression stabilities of the reference genes were analysed using geNorm, BestKeeper and NormFinder software (*Vandesompele et al., 2002*; *Andersen, Jensen & Orntoft, 2004*; *Pfaffl et al., 2004*). All the software tools were performed according to their manuals. The average Cq value was calculated from three biological and three technical replicates. To normalize the differences in the amounts of mRNA from other genes, the amount of *SpGAPDH* mRNA was determined for each sample. The relative expression level of S*pPDS* and *SpChlH* were analysed using the $2^{-\Delta\Delta CT}$ method (*Livak & Schmittgen, 2001*). The error bars represent the ±SEs from three independent experiments. The data were analysed by ANOVA using SAS software.

To determine whether leaf photobleaching was induced by TRV2-*SpPDS* and whether the yellow-leaf phenotype was induced by TRV2-*SpChlH* in *S. pseudocapsicum*, TRV RNA1 and RNA2 were detected using the following two primer sets: (1) pTRV1-F/pTRV1-R and (2) pTRV2-F/pTRV2-R. The primers were designed based on the TRV1 RNA1 sequence (GenBank ID AF406990) and the TRV2 RNA2 sequence (GenBank ID 406991). The PCR procedures were as follows: (1) 95 °C for 5 min, (2) 35 cycles of 95 °C for 30 s, 55 °C for 30 s, and 72 °C for 1 min, and (3) maintenance at 72 °C for 2 min.

## Construction of the pTRV2-*SpPDS* and pTRV2-*SpChlH* vectors

To generate the pTRV2-*SpPDS* vector, the inserted *SpPDS* fragment was amplified using the PDS_insert_EcoRI-F and PDS_insert_EcoRI-R primers, and the pTRV2- *SpChlH* vector was generated by amplifying the inserted *SpChlH* fragment using the ChlH_insert_SmaI-F and ChlH_insert_SmaI-R primers. These primers were designed according to the manufacturer's instructions for a one-step seamless cloning kit (Ju Hua Tai Ke, China). The pTRV2 vector was then digested with *EcoRI* and *SmaI* (Thermo Fisher Scientific Inc., USA). After gel extraction, the inserted *SpPDS* fragment containing the *EcoRI* site adapter and the inserted *SpChlH* fragment containing the *SmaI* site adapter were ligated into the *EcoRI*-digested TRV2 vector and the *SmaI*-digested TRV2 vector, respectively, using a one-step seamless cloning kit (Ju Hua Tai Ke, China), and the resulting vectors were then transformed into *E. coli* DH5a cells (TransGen, China). The presence of the *PDS*-containing and *ChlH*-containing inserts was confirmed by PCR using the TRV2_insert_yz-F and TRV2_insert_yz-R primers through detection of the corresponding pTRV2 multiple cloning sites (MCS). The expected size of the PCR product from the empty TRV2 vector was 338 bp, whereas a 909-bp product indicated that the *PDS*-containing fragment of interest was inserted into the TRV2 vector, and a 809-bp product indicated that the *ChlH*-containing fragment of interest was inserted into the TRV2 vector. The plasmid was further sequenced to verify the correct insertion of the fragment. Finally, the successfully constructed vector was transformed into Agrobacterium strain GV3101 or LBA4404 using the freeze-thaw method, as previously reported (*Jyothishwaran et al., 2007*).

## Agrobacterium inoculation and optimization of TRV-based VIGS conditions in *S. pseudocapsicum*

For the VIGS assay, the Agrobacterium strains GV3101 and LBA4404 containing the TRV-VIGS vector were cultured as described by *Tian et al. (2013)* with slight modifications. The Agrobacterium strains GV3101 and LBA4404 were harvested, resuspended in infiltration media (20 g/L sucrose, 5 g/L MS, 10 mM MES, and 200 mM acetosyringone) and adjusted to an $OD_{600}$ of 1.0. A mixture of Agrobacterium cultures containing pTRV1 and pTRV2 (or its derivatives) was placed in the dark at room temperature for 3 h before inoculation. Seedlings, seeds and sprouts were then infiltrated with Agrobacterium cultures containing pTRV1 and pTRV2 (or pTRV2 derivatives) (1:1, $OD_{600} = 1.0$) using three Agrobacterium inoculation methods. The detailed methods are described below:

Leaf syringe-infiltration method: When a seedling had developed two cotyledons, the underside of the cotyledons was rubbed gently with a 10-μL tip and then infiltrated with *Agrobacterium* inocula using a 1-mL needleless syringe.

Sprout vacuum-infiltration method: Sprouts that were 0.5–1 cm in length were submerged into agro-inocula in a beaker within a desiccator, pulled by a vacuum pump until the pressure reached 0.07 kPa, and maintained for 1 min.

Seed vacuum-infiltration method: Seeds were treated using the same protocol described for the sprout vacuum-infiltration method.

In the temperature optimization experiment, seedlings were treated with Agrobacterium strain GV3101 containing TRV2-*SpChlH* using the leaf syringe-infiltration method and

then grown at 18 °C, 25 °C, or 30 °C. The number of seedlings subjected to each treatment was 50, and each experiment was repeated three times.

## RESULTS

### Cloning and sequence analysis of *S. pseudocapsicum* orthologs for the reference genes and the *PDS* and *ChlH* genes

The reference genes and *PDS* and *ChlH* gene fragments were amplified by homology-based cloning as previously reported (*Zhong et al., 2014*). The results showed that the following fragments were obtained: approximately 600-bp *ACTIN* and *GAPDH* fragments, 200-bp *UBQ* fragment, 600-bp *PDS* fragment and 500-bp *ChlH* fragment (Fig. 1). A BLASTX search (https://blast.ncbi.nlm.nih.gov/Blast.cgi?PROGRAM=blastx&PAGE_TYPE=BlastSearch& LINK_LOC=blasthome) of the sequence results was conducted, and the search results showed that the *ACTIN*, *GAPDH*, *UBQ*, *PDS* and *ChlH* amplification fragments were derived from the corresponding genes. To obtain the longer *PDS* sequences, 3′-rapid amplification of cDNA ends was performed, and an approximately 761-bp partial *PDS* fragment was obtained (Fig. 1). In addition, an approximately 1,151-bp partial *PDS* fragment was obtained by sequence assembly based on the two above-mentioned *PDS* fragments using DNAMAN software (Lynnon Biosoft). Alignment of the *ACTIN*, *GAPDH*, *UBQ*, *PDS* and *ChlH* fragments of *C. annuum* and *S. pseudocapsicum* showed that these gene fragments shared high similarity between the two species; the *ACTIN* fragment (GenBank accession number: MG825852) in *S. pseudocapsicum* showed 99% similarity with the *ACTIN* gene fragment in *C. annuum*, whereas the *GAPDH* (GenBank accession number: MG825856), *UBQ* (GenBank accession number: MG825855), *PDS* (GenBank accession number: MG825854) and *ChlH* (GenBank accession number: M825853) gene fragments exhibited 88%, 100%, 95% and 97% similarity, respectively (Table 2). These results showed that the orthologous *ACTIN*, *GAPDH*, *UBQ*, *ChlH* and *PDS* gene fragments were cloned successfully in *S. pseudocapsicum*.

### Assessment of primer specificity and expression levels of candidate reference genes

For identification of RT-qPCR primer specificity, the PCR-amplified products were analysed by agarose gel electrophoresis and melting curve analyses. The three sets of RT-qPCR primer pairs generated single bands with the expected size (Fig. 2A). Additionally, melting curve analyses also yielded a single peak with no visible primer-dimer formation (Figs. 2B–2D). Thus, both analyses confirmed that the three reference gene primer pairs yielded specific amplification of the reference genes in *S. pseudocapsicum*. The correlation coefficients ($R^2$) of the standard curve were higher than 0.98 (Table 1), indicating good linear relationships among all the samples. The amplification efficiencies ranged from 97.7% for *ACTIN* to 105.7% for *UBQ*, suggesting that the reference gene primer pairs are suitable for further gene expression analyses (Table 1).

The expression levels of the three candidate reference genes are presented as raw Cq values. The results showed that the selected reference genes exhibited relatively wide expression abundance and variation. The mean Cq values for the reference genes ranged

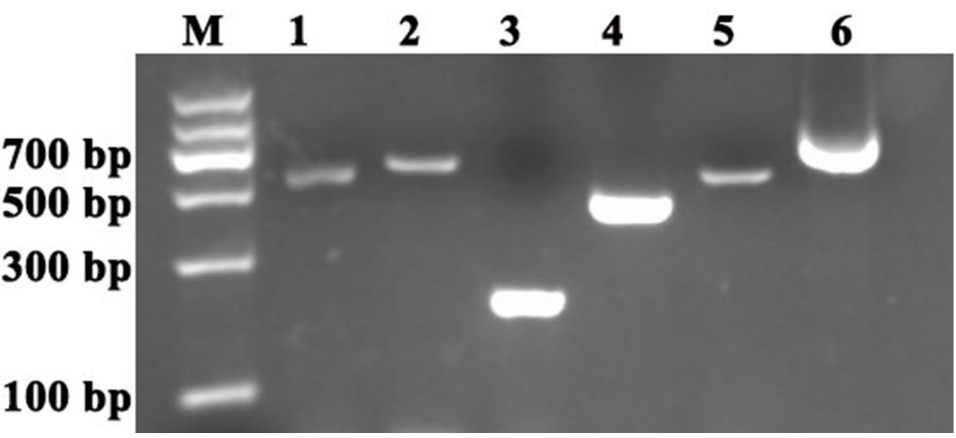

**Figure 1** **Cloning of the reference genes and *PDS* and *ChlH* gene fragments in *S. pseudocapsicum*.** Amplification of *SpACTIN*, *SpGAPDH*, *SpUBQ*, *SpPDS* and *SPChlH* from *S. pseudocapsicum* leaves. The primers for *SpACTIN*, *SpGAPDH* and *SpUBQ* were designed from conserved regions of *ACTIN*, *GAPDH* and *UBQ* based on the alignment of tomato, pepper, tobacco and potato *CDS* sequences. The primers for *SpPDS* and *SPChlH* were designed from conserved regions of *PDS* and *ChlH* based on the alignment of tomato, pepper, and tobacco *CDS* sequences. The cDNA from *S. pseudocapsicum* leaves was used as the template for the amplification of the corresponding PCR products. Lane 1, RT-PCR products of the *ACTIN* gene; Lane 2, RT-PCR product of the *GAPDH* gene; Lane 3, RT-PCR product of the *UBQ* gene; Lane 4, RT-PCR product of the *ChlH* gene; Lane 5, RT-PCR product of the *PDS* gene; and Lane 6, 3′RACE product of the *PDS* gene. The gene-specific primers (GSPs) for 3′RACE were designed based on the sequence of Lane 5 according to the manufacturer's instructions provided with the GeneRacer™ kit (Invitrogen, Carlsbad, CA, USA). All the primers are shown in Table S1.

**Table 2** **Candidate reference genes and *PDS* and *ChlH* gene fragments in *S. pseudocapsicum*.**

| Gene symbol | BlastX search | | | |
| --- | --- | --- | --- | --- |
| | Capsicum orthologue locus | Capsicum locus description | Similarity (*e*-value) | Identity (%) |
| *ACTIN* | XP_016543674.1 | Actin | 2e−139 | 99% |
| *GAPDH* | PHT69511.1 | Glyceraldehyde-3-phosphate dehydrogenase | 1e−123 | 88% |
| *UBQ* | AAR83898.1 | Ubiquitin-conjugating protein | 2e−47 | 100% |
| *PDS* | XP_016562403.1 | 15-cis-phytoene desaturase | 0.0 | 95% |
| *ChlH* | PHT80618.1 | Magnesium-chelatase subunit (ChlH) | 2e−101 | 97% |

from 31 (*SpACTIN*) to 21.3 (*SpUBQ*) (Fig. 3). None of the candidate reference genes was consistently expressed in the tested samples. Thus, the results showed that screening for suitable reference genes is critical for the analysis of gene expression in *S. pseudocapsicum*.

## Reference gene stability analysis

The stabilities of the reference genes were assessed using the geNorm, BestKeeper and NormFinder algorithms. The average expression stability value (*M* value) obtained using geNorm software was used to assess the gene expression stability. The M value of a suitable reference gene should be less than 1.5, and the gene with the lowest *M* value is considered to exhibit the greatest expression stability (*Vandesompele et al., 2002*). The results showed that the *M* values for the three candidate reference genes were below the geNorm threshold

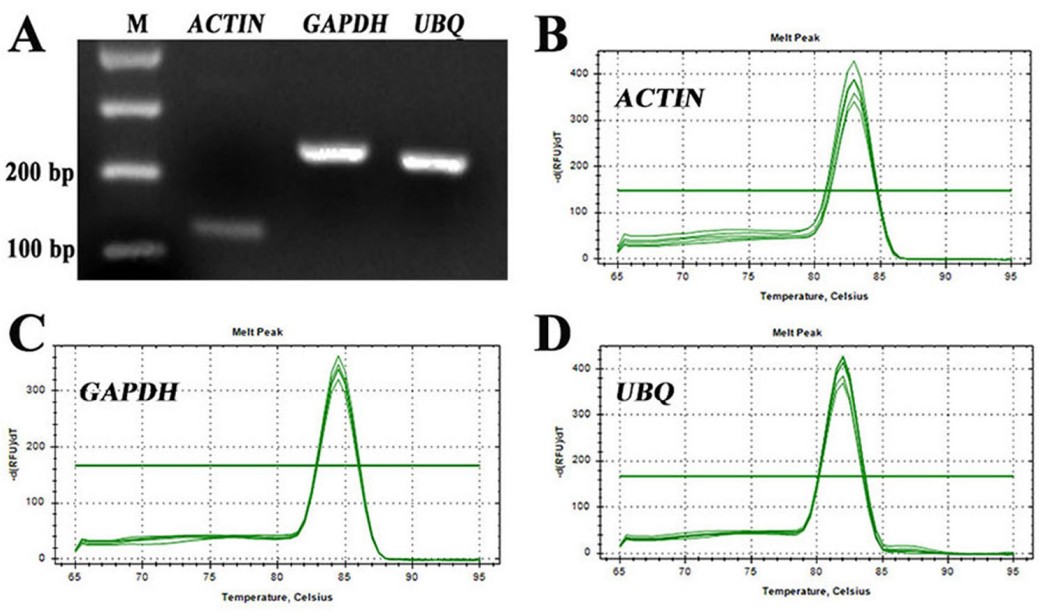

**Figure 2 Assessment of the specificity of the primers used for RT-qPCR amplification.** (A) Agarose gel electrophoresis showing the specific RT-PCR products of the expected size obtained for the three candidate genes. (B–D) Melting curves of the *ACTIN, GAPDH,* and *UBQ* reference genes showing single peaks (each was obtained from five-fold serial dilutions of pooled cDNA samples from leaves of *S. pseudocapsicum*).

of 1.5. In fact, the lowest *M* value was 0.865 (for *GAPDH*), whereas the highest *M* value was 1.109 (for *ACTIN*). The ranking of the *M* values was *GAPDH* (0.865) < *UBI* (1.053) < *ACTIN* (1.109) (Fig. 4A), which indicated that *GAPDH* and *ACTIN* were the most and least stable reference genes, respectively.

NormFinder software was used to further confirm the results obtained with the geNorm programme. Similar to geNorm software, the expression of the most stable gene was indicated by the lowest average expression stability value (*Andersen, Jensen & Orntoft, 2004*). The ranking of the stability values calculated using NormFinder was *GAPDH* (0.280) < *UBQ* (0.598) < *ACTIN* (0.671) (Fig. 4B). These results suggested that the genes showing the highest and lowest expression stability were *GAPDH* and *ACTIN*, respectively, which was in agreement with the results calculated using geNorm software.

BestKeeper software was also used to assess the expression stability of the reference candidate genes, and this assessment was performed by calculating the standard deviation (SD), the coefficient of variation (CV) and the Pearson correlation coefficient (r) from the raw Cq values (*Pfaffl et al., 2004*). Specifically, the lowest SD and CV values and the highest r value indicate the most stable reference gene. The ranking of the SD and CV values for the three reference candidates was *ACTIN* > *GAPDH* > *UBQ* (Table 3), indicating that *ACTIN* was the least stable reference gene. Although the SD and CV values for *UBQ* were lower than those for *GAPDH*, the r value for *GAPDH* (0.918) was higher than that obtained for

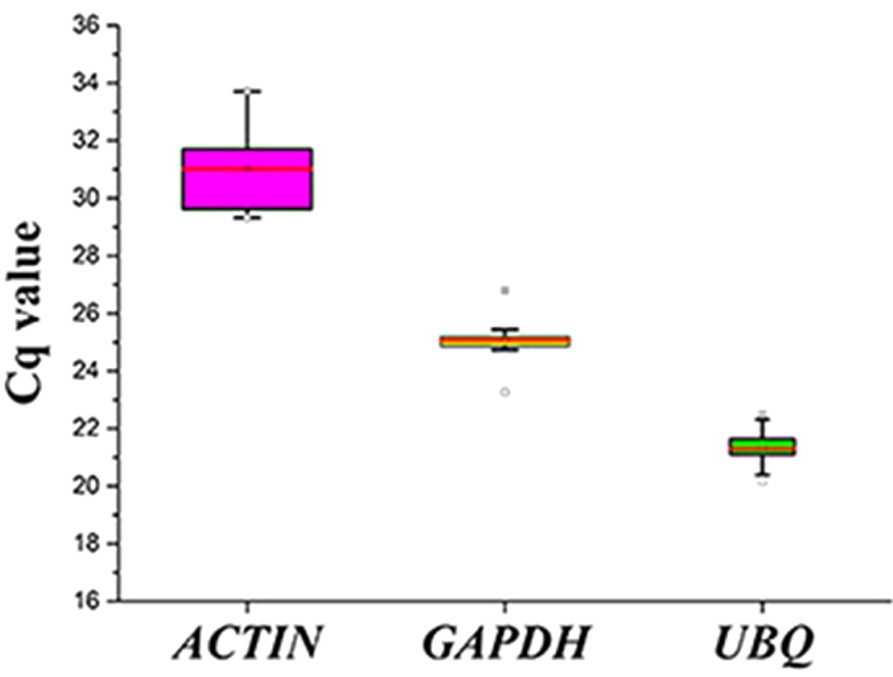

**Figure 3** **Cq values for the candidate reference genes in all the samples.** Expression levels of the three candidate reference genes in all the samples. The expression data are displayed as the Cq values for each reference gene in all the experimental samples ($n = 15$). A line across the box depicts the median values. The box indicates the 25th and 75th percentiles, and the whiskers represent the maximum and minimum values.

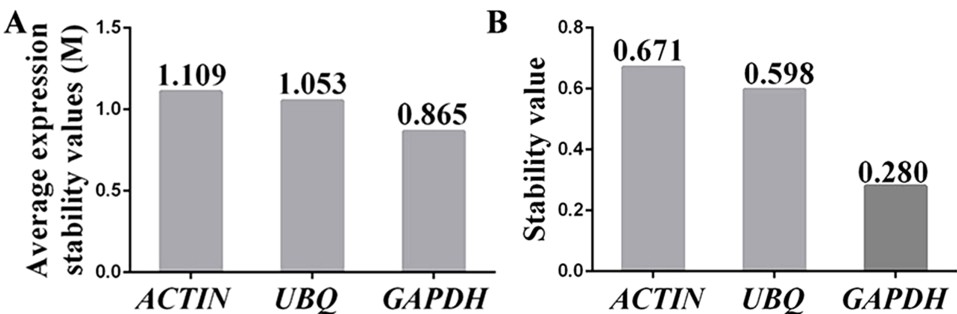

**Figure 4** **Assessment of the expression stability of the candidate genes calculated using (A) geNorm and (B) NormFinder.** (A) The expression stability (M) of each reference gene was calculated by geNorm software using 15 samples ($n = 15$). A lower M value indicates more stable gene expression. (B) The stability values of the reference genes were also calculated with NormFinder using 15 samples ($n = 15$), and a lower stability value indicates more stable gene expression. Each experiment was repeated three times.

**Table 3  Descriptive statistics for the three candidate reference genes calculated by BestKeeper.**

| Gene name | Standard deviation (SD) | Coefficient of variation (CV) | Pearson correlation coefficient (r) |
|---|---|---|---|
| ACTIN | 1.11 | 3.59 | 0.985 |
| GAPDH | 0.59 | 2.34 | 0.918 |
| UBQ | 0.27 | 1.29 | 0.832 |

UBQ (0.832). Thus, GAPDH was selected as the most suitable reference gene according to BestKeeper software.

Based on the results obtained using geNorm, NormFinder and BestKeeper software, GAPDH was selected as the most stable reference gene for detecting the expression levels of PDS and ChlH in the following experiments.

## Verification of the construction of the TRV2-*SpPDS* and TRV2-*SpChlH* vectors

To investigate the accuracy of the construction of TRV2-*SpPDS* and *TRV2-SpChlH*, PCR verification was performed using TRV2-*SpPDS* and *TRV2-SpChlH* as the template, respectively. A band with the expected size of approximately 909 bp was obtained in lane 2 (using TRV2-*SpPDS* as template), and a 809-bp band was obtained in lane 3 (using TRV2-*SpChlH* as template) (Fig. 5). These findings suggested that the PDS and ChlH fragments were introduced into the TRV2 vector, respectively. To further verify the results, we sequenced the TRV2-*SpPDS* and TRV2-*SpChlH* plasmids. Alignment of the sequencing results of TRV2-*SpPDS* and the inserted PDS fragment showed that the two sequences shared 100% similarity (Fig. S1), and the sequencing results of TRV2-Sp*ChlH* and the inserted ChlH fragment were similar to those obtained for TRV2-*SpPDS* (Fig. S2). These results demonstrate that the TRV2-*SpPDS* and TRV2-*SpChlH* vectors were accurately constructed.

## Silencing of the *PDS* and *ChlH* genes in *S. pseudocapsicum* seedlings

To investigate whether the TRV2-*SpPDS* and TRV2-*SpChlH* vectors could silence the corresponding endogenous PDS and ChlH genes in S. pseudocapsicum, the TRV2-*SpPDS* and TRV2-*ChlH* vectors were transformed into Agrobacterium strain GV3101, and the resulting strains were then inoculated into cotyledons using the leaf syringe-infiltration method. All the treated seedlings survived, indicating that the method was suitable for the tested seedlings. At 15 days post-infiltration (dpi), the photobleaching phenomenon was observed in the newly developed leaves obtained from the seedlings treated with TRV2-*SpPDS* but not in those obtained from the control and mock-treated seedlings (Fig. 6C, Table 4). Meanwhile, the yellow-leaf phenotype was detected in the newly developed leaves collected at 14 dpi from the seedlings treated with TRV2-*SpChlH* (Fig. 6K, Table 4). Moreover, the next newly developed leaves in the seedlings infiltrated with the TRV2-*SpPDS* and TRV2-*SpChlH* vectors also exhibited the photobleaching and yellow-leaf phenotypes, respectively (Figs. 6G and 6O). Additionally, we found that the silencing phenotypes

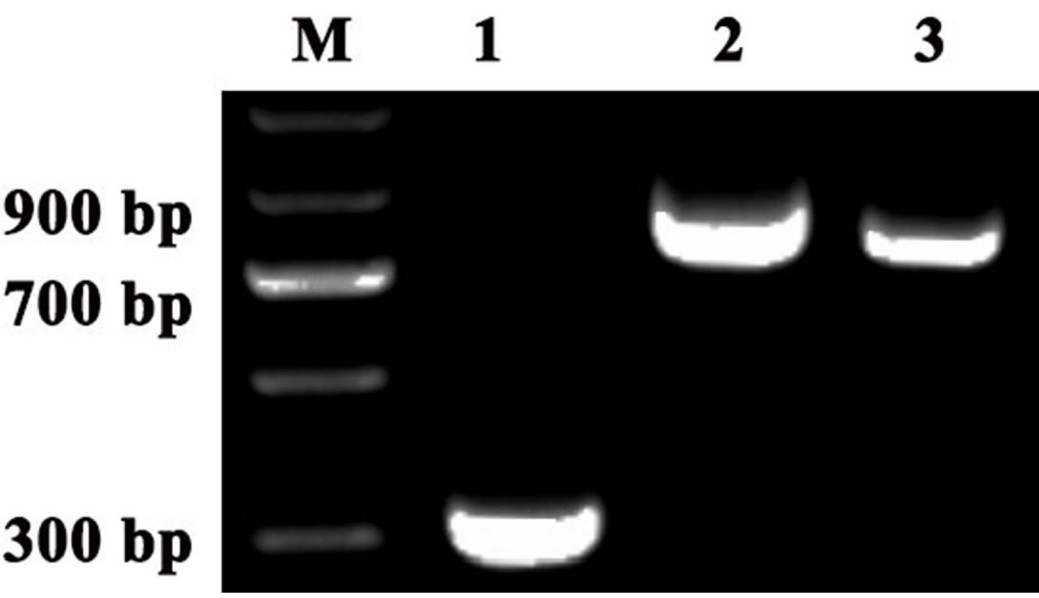

**Figure 5** **Detection of TRV2 and the TRV2-S*pPDS* and TRV2-*SpChlH* plasmids.** RT-PCR was performed with locus-specific primers to detect the multiple cloning sites (MCS) using different vectors as the template. Agarose gel electrophoresis showed specific RT-PCR products of the expected size for the different vectors. Lane M, marker; Lane 1, TRV2; Lane 2, TRV2-*SpPDS* vector; and Lane 3, TRV2-*SpChlH* vector. An approximately 338-bp band (Lane 1) indicated the empty TRV2 vector, whereas a 909-bp band (Lane 2) indicated that the TRV2-*SpPDS* vector was successfully constructed, and a 809-bp band (Lane 3) indicated that the TRV2-*SpChlH* vector was also successfully constructed.

**Table 4** **Effect of various infiltration methods on the silencing efficiency of *phytoene desaturase* (*PDS*) and *Mg-chelatase H subunit* (*ChlH*) in S. pseudocapsicum.**

| Agro-inoculation method | PDS | | | ChlH | | |
| --- | --- | --- | --- | --- | --- | --- |
| | NTS | NSP | DFI | NTS | NSP | DFI |
| Seed vacuum-infiltration | 50 | 0 | – | 50 | 0 | – |
| Leaf syringe-infiltration | 50 | 25 | 15 | 50 | 26 | 14 |
| Sprout vacuum-infiltration | 50 | 4 | 13 | 50 | 5 | 12 |

**Notes.**
NTS, number of treated seedlings; NSP, number of seedlings showing a silenced phenotype; DFI, first day after infiltration that the phenotype appeared.

obtained with the TRV2-*SpPDS* and TRV2-*SpChlH* vectors occurred along the leaf vein at the early silencing stage (Figs. 6D, 6H, 6L and 6P).

To investigate whether the photobleaching and yellow-leaf phenotypes were induced by the silencing of the corresponding endogenous *PDS* and *ChlH* genes, the *PDS* and *ChlH* expression levels in various VIGS-treated seedlings were detected by RT-qPCR using *GAPDH* as the reference gene. The *GAPDH* band was detected in all the samples, indicating that the cDNA from all the samples was of sufficient quality and could be used in a subsequent experiment (Fig. 7A). The RT-qPCR results showed that the *PDS* and *ChlH* expression levels were significantly reduced in the silenced seedling leaves compared with the leaves of the control and mock-treated seedlings. Moreover, the *PDS* and *ChlH*

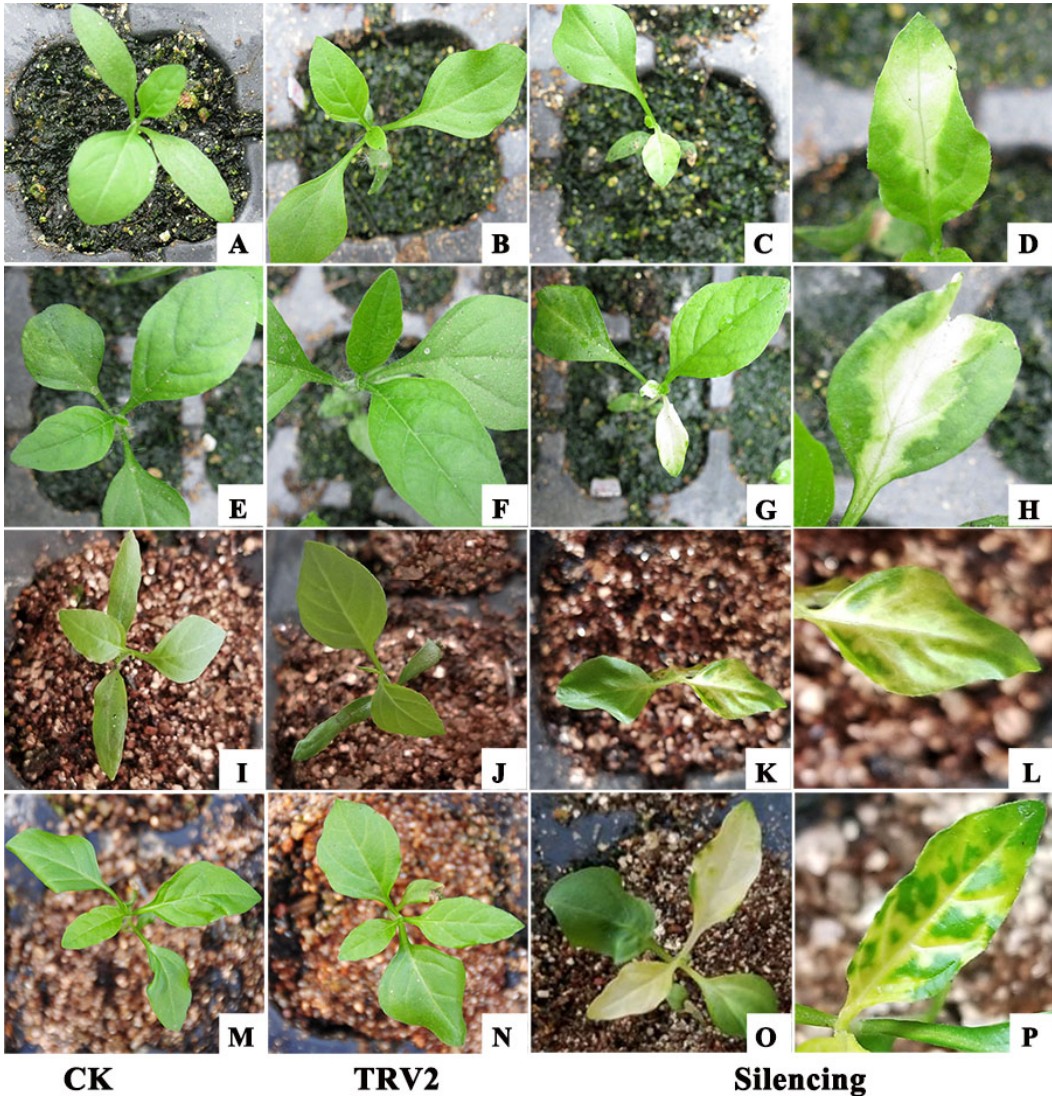

**Figure 6** **Establishment of a TRV-mediated VIGS protocol for *S. pseudocapsicum* using the leaf syringe-infiltration method.** All the TRV2 vectors were transformed into Agrobacterium strain GV3101 prior to its use for the treatment of *S. pseudocapsicum* seedlings. (A, E, I and M) CK, untreated *S. pseudocapsicum* seedlings. (B, F, J and N) TRV2 (mock), *S. pseudocapsicum* seedlings treated with TRV2. (C and G) Silenced *S. pseudocapsicum* seedlings treated with TRV2-*SpPDS*. The photobleached leaf phenotype observed in the *PDS*-silenced seedlings is shown in the first (C) and second (G) newly developed leaves. (K and O) Silenced *S. pseudocapsicum* seedlings treated with TRV2-*SpChlH*. The yellow-leaf phenotype observed in the *ChlH*-silenced seedlings is shown in the first (K) and second (O) newly developed leaves. (D, H, L and P) Silenced leaflets showing signs of the silencing phenotype along the vascular system. (D and H) Photobleached leaves of *PDS*-silenced seedlings. (L and P) Yellow leaves of *ChlH*-silenced seedlings. Photo credit: Hua Xu.

expression levels in the mock-infected and control leaves were similar (Figs. 7C, 7D). The above results indicated that the photobleaching and yellow-leaf phenotypes were initiated by *PDS* and *ChlH* gene silencing, respectively. Additionally, to determine whether the observed *PDS* and *ChlH* gene silencing was due to the presence of the TRV viral vector,

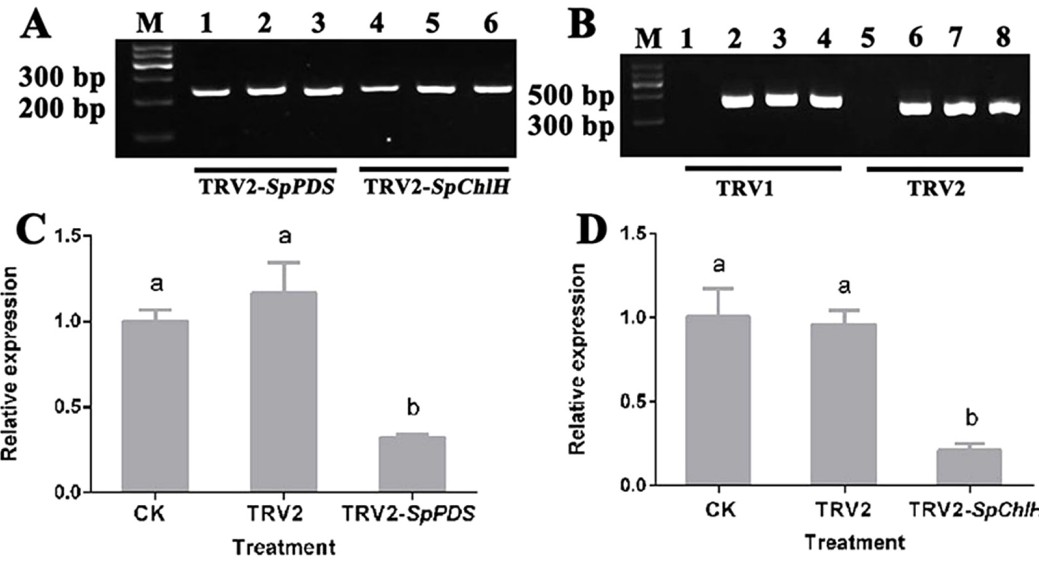

**Figure 7 Detection of the *SpPDS* and *SpChlH* expression levels and TRV transcripts after VIGS.** (A) RT-PCR detection of the *SpGAPDH* expression levels in leaves after various treatments. Lane M, marker; Lanes 1 and 4, untreated; Lanes 2 and 5, treated with TRV2; Lane 3, treated with TRV2-*SpPDS*; and Lane 6, treated with TRV2-*SpChlH*. (B) RT-PCR detection of TRV transcripts in leaves after various treatments. Lane M, marker; Lanes 1 and 5, untreated; lanes 2 and 6, treated with TRV2; Lanes 3 and 7, treated with TRV2-*SpPDS*; Lanes 4 and 8, treated with TRV2-*SpChlH*; Lanes 1, 2, 3 and 4, detection of TRV1; and Lanes 5, 6, 7 and 8, detection of TRV2. The second newly developed leaves from untreated and mock-treated *S. pseudocapsicum* seedlings were compared with leaves from TRV2-*SpPDS* or TRV2-*SpChlH*-treated seedlings. RT-PCR was performed with locus-specific primers for the reference gene *SpGAPDH* and the viral transcripts TRV1/TRV2. (C–D) *SpPDS* (C) and *SpChlH* (D) expression levels in the leaves after VIGS treatments. The expression levels of *SpPDS* and *SpChlH* were quantified via real-time RT-PCR in samples from leaves exhibiting the silenced phenotype and CK and mock-treated leaves at the two-true-leaf stage. *SpGAPDH* was used as an internal control for normalization of the PCR efficiency, and the *SpPDS* and *SpChlH* expression values of the CK were set to 1. The error bars represent the ±SE of three independent experiments. Different letters indicate significant differences at $P \leq 0.05$.

the TRV1 and TRV2 fragments in the infected and uninfected seedlings were assessed, and both the infected and the mock-inoculated seedlings showed the presence of the TRV1 and TRV2 bands (Fig. 7B), suggesting that silencing of the *PDS* and *ChlH* genes was induced by the presence of the TRV viral vector. The above-mentioned results indicated that TRV2-*SpPDS* could induce photobleaching by silencing of the endogenous *PDS* gene and that TRV2-*SpChlH* could induce the yellow-leaf phenotype by silencing of the endogenous *ChlH* gene in *S. pseudocapsicum*.

## Effect of various infiltration methods on silencing efficiency

To test whether the infiltration method could affect the silencing efficiency of the VIGS system in *S. pseudocapsicum*, the effects of various inoculation methods on the silencing efficiency were compared. The seed vacuum-infiltration method failed to induce any silencing phenotypes (Table 4), and no TRV transcript was detected in the leaves of *S. pseudocapsicum* (data not shown). The sprout vacuum-infiltration method could induce *PDS* and *ChlH* gene silencing with a silencing efficiency of approximately 10% at 12–13 dpi,

**Table 5** Effect of Agrobacterium strains on the silencing efficiency of *phytoene desaturase* (*PDS*) and *Mg-chelatase H subunit* (*ChlH*) in *S. pseudocapsicum.*

| Agrobacterium strain | Agro-inoculation method | Silencing efficiency | |
|---|---|---|---|
| | | *PDS* | *ChlH* |
| GV3101 | Leaf syringe-infiltration | 50% | 52% |
| | Sprout vacuum-infiltration | 8% | 10% |
| LBA4404 | Leaf syringe-infiltration | 44% | 40% |
| | Sprout vacuum-infiltration | 4% | 6% |

which is relatively low. In contrast, using the leaf syringe-infiltration method, the infected phenotypes were first detected at 14–15 dpi, and the TRV2-*SpPDS* and TRV2-*SpChlH* vectors induced the respective phenotypes in approximately 50% of treated seedlings (Table 4). The results indicated that the leaf syringe-infiltration method was the most effective for inducing the silencing phenotype in *S. pseudocapsicum*.

## Effect of the Agrobacterium strain on the silencing efficiency

To investigate whether the Agrobacterium strains used to introduce the VIGS vectors affected the silencing efficiency, two Agrobacterium strains, GV3101 and LBA4404, were used to introduce the TRV2-*SpPDS* and TRV2-*SpChlH* vectors, respectively. The results showed that both Agrobacterium strains could introduce the VIGS vectors used for silencing of the gene of interest, but the silencing efficiency of *PDS* and *ChlH* induced by GV3101 was higher than that induced by LBA4404, regardless of whether the leaf syringe-infiltration or the sprout vacuum-infiltration method was used (Table 5). The results indicated that GV3101 was better for the introduction of the VIGS vectors into *S. pseudocapsicum* than LBA4404.

## Effect of the growth temperature of agro-inoculated plants on the silencing efficiency

To investigate whether the growth temperature of the agro-inoculated plants affected the silencing efficiency, we compared the *ChlH* silencing efficiency under different growth temperatures (18 °C, 25 °C and 30 °C) after agro-inoculation using the leaf syringe-infiltration method. Analysis of the agro-inoculated seedlings showed that 52% of the seedlings that grew at 25 °C developed the yellow-leaf phenotype at 14 dpi, 12% of the seedlings that grew at 18 °C developed a yellow-leaf phenotype at 22 dpi, and 4% of the seedlings that grew at 30 °C developed the silencing phenotype at 28 dpi (Table 6). Thus, both higher and lower growth temperatures decreased the silencing efficiency and delayed the time from infection to obvious phenotype detection. The results indicated that the speed of the response and the efficiency of infection were optimal at 25 °C.

## DISCUSSION

Gene function studies promote increased understanding of the molecular mechanisms in plants, and to date, few gene function studies have been performed in *S. pseudocapsicum*. VIGS is a recently developed powerful genetic tool for characterizing the function of plant

**Table 6  Effect of temperature on the silencing efficiency of the *Mg-chelatase H subunit* (*ChlH*) in *S. pseudocapsicum*.**

| Growth temperature (°C) | % of plants with the yellow-leaf phenotype | Days to phenotype detection |
| --- | --- | --- |
| 18 | 12 | 22 |
| 25 | 52 | 14 |
| 30 | 4 | 28 |

genes (*Burch-Smith et al., 2004*). *PDS* and *ChlH* have been commonly used as indicator genes in VIGS systems because their resulting silencing phenotypes can be easily scored (*Cunningham & Gantt, 1998*; *Hiriart et al., 2002*; *Liu et al., 2012*). In the present study, we established that TRV-based VIGS could be applied to unravel the functions of the *PDS* and *ChlH* genes in *S. pseudocapsicum*. Systemic viral infection was essential for the VIGS system. Our results also suggested that newly developed leaves obtained after VIGS application exhibited the silencing phenotype, indicating establishment of systemic TRV viral infection in *S. pseudocapsicum* (Figs. 6C, 6G, 6K and 6O). Additionally, the results showed that both silencing phenotypes occurred along the leaf vein (Figs. 6D, 6H, 6L and 6P), which agrees with the results that viral propagation and the silenced systemic response occurred mainly along the vascular bundle system (*Wege et al., 2007*). To the best of our knowledge, this study provides the first demonstration of a successful application of TRV-based VIGS for the identification of gene function in *S. pseudocapsicum*.

We also compared several parameters that are likely to affect the silencing efficiency in *S. pseudocapsicum*, including the Agrobacterium inoculation method, the Agrobacterium strain and the growth temperature after Agrobacterium infiltration. In some previous studies, the vacuum-infiltration method was found to be more effective than other infiltration methods (*Deng et al., 2012*; *Liu et al., 2014*). However, our results suggested that the leaf syringe-infiltration method is more effective than vacuum-infiltration in *S. pseudocapsicum* (Table 4). This difference might have been observed due to the high hardness of the seed coat of *S. pseudocapsicum*, which might have not allowed infection with the virus vector.

It is well known that susceptibility to infection by Agrobacterium varies among different plant species and cultivars. For example, in Arabidopsis, agro-inoculation with LBA4404 does not give rise to a silencing phenotype, whereas the GV3101 strain results in highly efficient VIGS (*Cai et al., 2006*). However, in *Gossypium barbadense*, agro-infiltration with the GV3101 and LBA4404 strains resulted in efficient VIGS (*Pang et al., 2013*). Here, we evaluated the silencing effect of two strains of Agrobacterium, GV3101 and LBA4404, in *S. pseudocapsicum*. The results showed that both GV3101 and LBA4404 could introduce the VIGS vector to produce the silencing phenotype (Table 5), which agrees with the results obtained in *G. barbadense* (*Pang et al., 2013*). *Pang et al. (2013)* found that the two Agrobacterium strains GV3101 and LBA4404 yielded no significant difference in the VIGS efficiency (*Pang et al., 2013*), but in this study, the silencing efficiency in *S. pseudocapsicum* obtained with GV3101 was slightly higher than that obtained by LBA4404, which might be due to the differences between the species.

The growth temperature conditions after inoculation have a profound effect on silencing efficiency (*Burch-Smith et al., 2004*). Different plant species require different temperatures for producing a good silencing phenotype after VIGS. For example, in tomato, the optimal silencing phenotype is obtained with a growth temperature of 22 °C after TRV-based VIGS (*Jiang et al., 2008*), whereas temperatures of approximately 25 °C are desirable for *N. benthamiana* (*Burch-Smith et al., 2004*). In the present study, we also found that the growth temperature after inoculation affected the silencing efficiency. A decreased silencing efficiency was obtained when the seedlings grew at lower (18 °C) or higher (30 °C) temperatures, and the most suitable growth temperature for obtaining a high silencing efficiency in *S. pseudocapsicum* was 25 °C (Table 6).

The growth stage of the agro-inoculated plants can also affect the gene silencing efficiency (*Burch-Smith et al., 2004*; *Deng et al., 2012*). In addition to at the cotyledon stage, we also inoculated the seedlings at the two-true-leaf and four-true-leaf stages with TRV2-*SpPDS* using the leaf syringe-infiltration method, which was predicted to initiate the photobleaching phenotype in leaves. Unfortunately, the *SpPDS* silencing phenomenon was not observed in the seedlings' leaves (data not shown), indicating that the optimal VIGS inoculation stage in *S. pseudocapsicum* was the cotyledon stage.

RT-qPCR is an important technique for assessing gene expression levels. The reliability of RT-qPCR results is highly dependent on the suitability of the reference gene. It is thus necessary to screen for a suitable reference gene prior to the assessment of gene expression levels. To date, suitable reference genes for *S. pseudocapsicum* have not been well characterized. Thus, to accurately assess the gene expression levels in seedlings in which a silencing phenotype was induced with TRV2-*SpPDS* and TRV2-*SpChlH*, we screened for the most stable reference gene in this study. We compared three potential reference genes (*ACTIN, GAPDH, and UBQ*) for *S. pseudocapsicum* across five samples collected from leaves at different developmental stages and various tissues. The *GAPDH* gene was identified as the most stable reference gene for assessing gene expression levels in this study. To the best of our knowledge, this study constitutes the first screening for a stable reference gene in *S. pseudocapsicum* and lays the foundation for the characterization of gene expression levels in *S. pseudocapsicum*. Previous studies have shown that the expression levels of the reference genes varied under different experimental conditions (*Sinha et al., 2015*; *Li et al., 2016*; Sudhakar *Reddy et al., 2016*). Thus, the most suitable reference gene, which was identified as *GAPDH* in this study, should be subjected to further screening in *S pseudocapsicum* under other experiment conditions.

The rankings of the three reference genes obtained in the present study showed some discrepancies. For example, *GAPDH* was identified as the most stable reference gene using geNorm and NormFinder software, but the ranking order obtained for *UBQ* was higher than that found for *GAPDH* using BestKeeper. This difference might be due to the different statistical models used in the various algorithms. Differences in the ranking orders obtained with these algorithms have also been observed in other studies (*Tong et al., 2009*; *Reddy et al., 2016*; *Xu et al., 2017*).

## CONCLUSIONS

In summary, we established a TRV-based VIGS system that can successfully induce photobleaching and yellow-leaf phenotypes by silencing *SpPDS* and *SpChlH*, respectively. Using the optimal parameters, including Agrobacterium strain GV3101, the leaf syringe-infiltration method, and a growth temperature for the agro-inoculated plants of 25 °C, the silencing efficiency can reach approximately 50%. This established system will facilitate the future characterization of gene functions in *S. pseudocapsicum*. Additionally, the present study constitutes the first cloning and screening of reference genes and thus lays the foundation for gene expression analysis in *S. pseudocapsicum*. These results provide a better understanding of the molecular and physiological mechanisms that regulate *S. pseudocapsicum* and its associated traits.

## ACKNOWLEDGEMENTS

We are grateful to Dr. Xiaomei Su (Institute of Vegetables and Flowers, Chinese Academy of Agricultural Sciences, Beijing, China) for kindly offering pTRV1 and pTRV2. This research was conducted at the Key Laboratory of Biology and Genetic Improvement of Horticultural Crops, Ministry of Agriculture, China.

### Funding

This study was supported by the National Natural Science Foundation of China (31272205, 31672196), the Fundamental Research Funds for Central Non-profit Scientific Institutions, and the Science and Technology Innovation Program of the Chinese Academy of Agricultural Sciences. The funders had no role in study design, data collection and analysis, decision to publish, or preparation of the manuscript.

### Grant Disclosures

The following grant information was disclosed by the authors:
National Natural Science Foundation of China: 31272205, 31672196.
The Fundamental Research Funds for Central Non-profit Scientific Institutions.
Science and Technology Innovation Program of the Chinese Academy of Agricultural Sciences.

### Competing Interests

The authors declare there are no competing interests.

### Author Contributions

- Hua Xu conceived and designed the experiments, performed the experiments, prepared figures and/or tables, approved the final draft.
- Leifeng Xu performed the experiments, prepared figures and/or tables, authored or reviewed drafts of the paper, approved the final draft.

- Panpan Yang performed the experiments, analyzed the data, authored or reviewed drafts of the paper, approved the final draft.
- Yuwei Cao performed the experiments, approved the final draft.
- Yuchao Tang performed the experiments, contributed reagents/materials/analysis tools, approved the final draft.
- Guoren He analyzed the data, approved the final draft.
- Suxia Yuan analyzed the data.
- Jun Ming conceived and designed the experiments, analyzed the data, authored or reviewed drafts of the paper, approved the final draft.

### Data Availability

MG825852, MG825853, MG825854, MG825855, MG825856.

### Supplemental Information

Supplemental information for this article can be found online at http://dx.doi.org/10.7717/peerj.4424#supplemental-information.

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
