# Peer review of "Tobacco rattle virus-induced PHYTOENE DESATURASE (PDS) and Mg-chelatase H subunit (ChlH) gene silencing in Solanum pseudocapsicum L"

_PeerJ, doi:10.7717/peerj.4424_

## Round 0.1 · original submission · Major Revisions

As suggested by both the reviewers, the writing needs to be improved. The silencing efficiency of 25% is not optimal for a general use of VIGS in this plant sp and hence it is better to try different Agro strains, temperature, and inoculation conditions to optimize silencing method to increase the efficiency. As pointed out by the Reviewer #2, authors should use at least additional genes to show that the TRV-based silencing generally works in this plant species not just PDS.

Reviewer 1 ·

Basic reporting

This manuscript needs through English language editing.

Experimental design

• Text at many places are ambiguous e.g., L90-92, L133-34 also check overall for the grammatical errors.
• Proper terminology should be used e.g., L88, steam sterilized should be replaced with autoclaved.
• Age of the plant should also be mentioned in L90
• Sampling at L94 is not clear. Five different plants and three biological replicates?
• L97- what tissue was used?
• Name of the supplier is needed in L99
• Homology-based cloning section can be expanded with the mention of closely related species used and gene IDs used for the cloning. Mention if degenerate primers were used. Also, mention the template used for PCR and number of cycles.
• Some of the headings are not conclusive e.g. L130
• L131, where was PDS inserted?
• L131-137 needs improvement for better clarity
• Reference is needed in L110, L143
• Mention of kit or method is needed in L112

Validity of the findings

• Overall clarity in writing is needed e.g., L162-64. Some of the headings need improvement for the better conclusion e.g., L161, L220. Check for the capital and small letter placement
• Mention the name of the species used for homology in L162
• I don’t find any figure for 3’RACE result

Additional comments

Abstract
Information is sufficient but improvement in writing/language is needed for better clarity and readability.

Introduction
Information and citations look conclusive but writing requires improvement for better understanding e.g., line 17.

Figures
Figure 1: legend can be expanded with details of methods used and with details of the template used and a number of cycles.
Figure 2- details are needed for each figure like if A) is RT-PCR and B) is qRT-PCR. Needs better sentence framing for the clarity. Mention number of replicates for RT-qPCR
Figure 3- expand the legend with tissue sample used for the experiment and number of replicates used
Figure 4- expand the legend with details of input values and number of replicates
Figure 5- what is 1,1 and 2,2 in the figure? Expand the legend
Figure 6- check for the spelling mistakes in legend. Legend is confusing, what is the difference in C and D figures. Expand the legends with the methodology details
Figure 7 can be merged with figure 8
Figure 8- labels on figure do not match with the legend for 8A. also mention the age of tissue used and a number of replicates for qRT-PCR.

Other points that need attention
• How was the consistency of silencing in plant lifespan?
• Was silencing in other tissue parts of the plant found?
• Did silencing inherit to the next generation?

Reviewer 2 ·

Basic reporting

English writing should be improved. L45, L47 the two viruses name should be italic. L61 A should be a. L63 RT-qPCR. L76 A blank before Thus. L80 correct provide to provided. L83-L84 delete the last sentence. L121 delete five. L132 -133 Eco should be italic. L137 DH5a should not be italic. L143 need reference.

Plant materials only cover the plant used in study. L90-95 should be introduced in each part of materials and methods.

L108 sequences of these genes should be attached.
L110 Primers should be as supplementary data.
L169 GenBank accession numbers should be given for these cloned genes.
L175-184 should be merged into next section. L182 delete the first the.
L261 The silencing rate is not clear. The authors should clear ow many plants were inoculated, how many plants were infected, and how many plants showed silencing phentotype.

Experimental design

The main purpose of this work is to report TRV could be used as VIGS vector in Solanum pseudocapsicum. However, the authors put much words on reference genes selection for RT-qPCR, though it is important for this plant. I suggest the authors to have more work to optimize the inoculation methods to improve silencing efficience, and try to silence one more gene for solid data.

Validity of the findings

I could not understand "The silencing rate was 26%". The silencing efficience should be shown in the result.

---

## Round 0.2 · accepted · Accept

The revised manuscript addressed most of the comments raised by the reviewers.